# Are Undernutrition and Obesity Associated with Post-Discharge Mortality and Re-Hospitalization after Hospitalization with Community-Acquired Pneumonia?

**DOI:** 10.3390/nu14224906

**Published:** 2022-11-19

**Authors:** Maria H. Hegelund, Camilla K. Ryrsø, Christian Ritz, Arnold M. Dungu, Adin Sejdic, Andreas V. Jensen, Nikita M. Hansen, Christian Mølgaard, Rikke Krogh-Madsen, Birgitte Lindegaard, Daniel Faurholt-Jepsen

**Affiliations:** 1Department of Pulmonary and Infectious Diseases, Copenhagen University Hospital—North Zealand, 3400 Hillerød, Denmark; 2Centre for Physical Activity Research, Copenhagen University Hospital—Rigshospitalet, 2100 Copenhagen, Denmark; 3The Research Department for Health and Morbidity in the Population, National Institute of Public Health, University of Southern Denmark, 1455 Copenhagen, Denmark; 4Department of Infectious Diseases, Copenhagen University Hospital—Rigshospitalet, 2100 Copenhagen, Denmark; 5Department of Nephrology, Copenhagen University Hospital—Rigshospitalet, 2100 Copenhagen, Denmark; 6Department of Nutrition, Exercise and Sports, University of Copenhagen, 1958 Frederiksberg, Denmark; 7Pediatric Nutrition Unit, Copenhagen University Hospital —Rigshospitalet, 2100 Copenhagen, Denmark; 8Department of Infectious Diseases, Copenhagen University Hospital—Amager/Hvidovre, 2650 Hvidovre, Denmark; 9Department of Clinical Medicine, University of Copenhagen, 2200 Copenhagen, Denmark

**Keywords:** community-acquired pneumonia, undernutrition, obesity, re-hospitalization, mortality

## Abstract

Undernutrition is associated with increased mortality after hospitalization with community-acquired pneumonia (CAP), whereas obesity is associated with decreased mortality in most studies. We aimed to determine whether undernutrition and obesity are associated with increased risk of re-hospitalization and post-discharge mortality after hospitalization. This study was nested within the Surviving Pneumonia cohort, which is a prospective cohort of adults hospitalized with CAP. Patients were categorized as undernourished, well-nourished, overweight, or obese. Undernutrition was based on diagnostic criteria by the European Society for Clinical Nutrition and Metabolism. Risk of mortality was investigated using multivariate logistic regression and re-hospitalization with competing risk Cox regression where death was the competing event. Compared to well-nourished patients, undernourished patients had a higher risk of 90-day (OR 3.0, 95% CI 1.0; 21.4) mortality, but a similar 30-day and 180-day mortality risk. Obese patients had a similar re-hospitalization and mortality risk as well-nourished patients. In conclusion, among patients with CAP, undernutrition was associated with increased risk of mortality. Undernourished patients are high-risk patients, and our results indicate that in-hospital screening of undernutrition should be implemented to identify patients at mortality risk. Studies are required to investigate whether nutritional therapy after hospitalization with CAP would improve survival.

## 1. Introduction

Community-acquired pneumonia (CAP) remains a leading cause of hospitalization and death worldwide [1,2]. Undernutrition has been associated with an increased risk of acquiring CAP [3], longer hospital stays [4,5], and mortality [6,7,8,9] after hospitalization with CAP. Among patients hospitalized with viral CAP, undernutrition has been associated with higher risk of in-hospital complications including in-hospital mortality, septic shock, myocardial infarction, thrombosis, and pulmonary embolism [10]. Across different medical specialties, undernutrition has also been associated with higher risk of complications (e.g., sepsis, respiratory failure, cardiac arrest, and cardiac failure), higher cost, and re-hospitalization [11].

Among adults hospitalized with CAP, up to 40% are undernourished [4,6,10,12], with prevalence being highly dependent on the population and definition used. Individuals with chronic diseases, such as cancer, chronic obstructive pulmonary disease (COPD), and cardiovascular diseases are at increased risk of unintentional weight loss [13,14,15] and poor CAP-related outcomes including mortality [16,17]. The underlying chronic condition is often connected with the increased risk of poor CAP-related outcomes instead of undernutrition itself [16]. On the other hand, obesity has been associated with decreased mortality and better outcomes, even in patients with chronic diseases such as cardiovascular diseases [18,19], and type 2 diabetes [20], as well as in patients hospitalized with CAP [21], which is a phenomenon called the “obesity paradox”.

Studies among undernourished CAP patients mainly categorize undernutrition based on BMI, as <18.5 kg/m^2^, according to recommendations by the World Health Organization [22]. Alternatively, a few studies use anthropometric measures (arm circumference and triceps skinfold) or level of serum albumin [23]. The European Society of Clinical Nutrition and Metabolism (ESPEN) has suggested a complementary definition of undernutrition that includes weight loss combined with either alternative cut-offs for low BMI according to age or cut-offs for low fat-free mass index (FFMI) according to sex [24]. Studies report inconsistent results concerning obesity as a protecting factor for death in CAP, and remarkably few studies have investigated whether obesity is associated with re-hospitalization. To fully understand the risk of undernutrition and the opportunities for prophylaxis, there is a need for a broader focus that considers essential nutritional aspects other than just BMI.

The aim of this study was to determine whether undernutrition and obesity are associated with increased risk of re-hospitalization and mortality after discharge from hospitalization with CAP.

## 2. Materials and Methods

### 2.1. Study Design, Setting, and Population

This study is nested within the Surviving Pneumonia Cohort (ClinicalTrials.gov: NCT03795662), which is an ongoing prospective cohort including adults (≥18 years) admitted with CAP to Copenhagen University Hospital, North Zealand, Denmark, from the 7 January 2019. For the current study, patients were enrolled until the 4 April 2022. Inclusion criteria were CAP, defined as a new infiltrate on chest X-ray or computed tomography scan and a minimum of one of the following symptoms: fever (≥38.0 °C), cough, pleuritic chest pain, dyspnea, or focal chest signs on auscultation. Exclusion criteria for the current study were participation in other research studies potentially affecting the outcome of interest (i.e., re-hospitalization and mortality) and insufficient data for nutritional categorization (i.e., no BMI, FFMI, and information on weight loss). Outcome variables were re-hospitalization and mortality from discharge as well as 30, 90, and 180 days after discharge.

### 2.2. Data Collection

Upon admission, information about socio-demography and vital parameters were collected through patient interviews and reviews of electronic medical records. Pneumonia severity was assessed using the CURB-65 score, which is a tool to stratify patients according to mortality risk upon hospital admission. The CURB-65 score includes the following five parameters: confusion, urea (>7mmol/L), respiratory rate ≥30/min), blood pressure (systolic <90 mm Hg or diastolic ≤60 mm Hg), and age (≥65 years). One point is provided by the presence of any of the parameters and is used to categorize pneumonia severity as mild (0–1), moderate (2), or severe (3–5) [25]. Prevalence of the most frequent comorbidities (COPD, cardiovascular diseases, diabetes mellitus, and cancer) was collected through electronic medical records, and the combined burden of comorbidities was assessed using the Charlson comorbidity index. The Charlson comorbidity index is a tool developed to predict mortality by weighing multiple comorbidities; this includes myocardial infarction, congestive heart failure, peripheral vascular disease, cerebrovascular disease, dementia, chronic pulmonary disease, rheumatologic disease, peptic ulcer disease, mild liver disease, diabetes with or without chronic complications, hemiplegia or paraplegia, renal disease, any malignancy, moderate or severe liver disease, metastatic solid tumor, and HIV/AIDS. Each condition provides a score of 1, 2, 3, or 6 depending on its associated risk of mortality [26].

#### 2.2.1. Information on Weight Loss

Information on weight loss was extracted from the Nutritional Risk Screening 2002 [27] and FRAIL scale [28] which are two questionnaires answered in collaboration with the patient within 48 h after admission. A weight loss of ≥5% within the previous three months was extracted from the Nutritional Risk Screening 2002, and a weight loss of ≥10% within the previous 12 months was extracted from the FRAIL scale. Information on weight loss was used as part of the categorization.

#### 2.2.2. Anthropometry and Fat-Free Mass Index

Anthropometry and fat-free mass were measured within 48 h after admission. Weight was measured to the nearest 0.1 kg on an electronic scale (Seca, Hamburg, Germany), whereas height was self-reported. BMI was calculated as weight (kg)/height (m^2^). Fat-free mass was assessed using bioelectrical impedance analysis (BioScan touch i8, Maltron International Ltd., Essex, United Kingdom), and FFMI was calculated as fat-free mass (kg)/height (m^2^).

#### 2.2.3. Nutritional Categorization

Patients were categorized into four groups according to their nutritional status as undernourished, well-nourished, overweight, or obese. The definition of undernutrition was based on the diagnostic criteria by ESPEN as (1) a BMI of <18.5 kg/m^2^ or (2) unintentional weight loss combined with either an alternative cut-off of BMI according to age or a low FFMI according to sex [24], as illustrated in Table 1. Significant weight loss was defined as ≥5% loss of body weight within the previous three months or ≥10% lost within the previous 12 months. Patients with a BMI between 18.5 kg/m^2^ and 24.9 kg/m^2^ without significant weight loss and low FFMI were categorized as well-nourished. Overweight and obese patients were defined as patients with a BMI of 25.0–29.9 kg/m^2^ and a BMI of ≥30 kg/m^2^ with or without prior weight loss.

#### 2.2.4. Laboratory Data

Venous blood samples were collected within 24 h after admission and analyzed at the local accredited Clinical Biochemistry Department for levels of C-reactive protein (CRP). CRP is a measure of systemic inflammation used as a supporting biomarker when diagnosing infection, as well as a response marker to monitor changes during treatment. We determined CRP levels of ≤3 mg/L to be within the normal range and levels of 3–10 mg/L to be slightly elevated.

#### 2.2.5. Prognostic Data during Admission and Outcome Measures

Information on length of hospital stay, admission to the ICU, re-hospitalization (30-day, 90-day, and 180-day), and mortality (30-day, 90-day, and 180-day) were collected through electronic medical records.

### 2.3. Data Analysis

Statistical analyses were carried out using STATA/IC version 17.0 (StataCorp LP, College Station, TX, USA). The distribution of all variables was evaluated, and skewed variables were summarized using median and interquartile range (IQR), whereas normally distributed variables were summarized using mean and standard deviation (SD). All missing data were imputed using multiple imputation through chained equations (MICE) [29,30]. Cox regression was used to compare the risk of a minimum of one re-hospitalization within 30, 90, and 180 days after discharge. Patients without a re-hospitalization within the considered time periods (i.e., between discharge and 30, 90, or 180 days after discharge) and patients who died after their first discharge (without a re-hospitalization) were censored. Moreover, to avoid bias, death was handled as a competing event in the modelling [31]. Logistic regression was used to compare mortality risk between discharge and 30, 90, and 180 days after discharge. In the main analyses, the well-nourished group was set as reference. Supplementary analyses were conducted with the overweight as reference group. Cox and logistic regression models both included adjustment for age, sex, and COPD, while Kaplan-Meier estimates were used to assess the probability of re-hospitalization and mortality. Results are reported as hazard ratios (HR) and odds ratios (OR) with corresponding 95% confidence intervals (CI) and *p*-value. *p*-values ≤ 0.05 were considered significant.

### 2.4. Ethical Statement

The Surviving Pneumonia Cohort was approved by the Scientific Ethics Committee at the Capital Region of Denmark (H-18024256), registered on ClinicalTrials.gov (NCT03795662), and conducted in accordance with the Declaration of Helsinki. Oral and written informed consent was obtained from all patients before enrolment.

## 3. Results

Of the 790 patients admitted with CAP, 323 (41%) were included in the current study. A total of 70 patients were excluded due to in-hospital mortality and withdrawal. Another 467 of the patients were ineligible due to participation in a physical training intervention and insufficient data on BMI, FFMI, and weight loss to categorize them into the appropriate nutritional group. Figure 1 illustrates the flow of study inclusion. No difference was found between eligible and ineligible patients regarding the following characteristics: age, sex, comorbidity index, length of hospital stay, CURB-65, ICU admission, CRP, re-hospitalization, or mortality (Appendix A).

## 4. Characteristics during Hospitalization and Outcome Characteristics

Characteristics during hospitalization and outcome characteristics are shown in Table 2 and Appendix A. Overall, the mean ± SD age was 69.8 ± 13.8 years. The prevalence of COPD was higher among undernourished patients at 58% compared to 33%, 30% and 26% in well-nourished, overweight, and obese patients, respectively. However, the difference between undernourished and well-nourished did not reach the significant threshold. Prevalence of cardiovascular diseases, cancer, and diabetes were similar between the groups (Appendix A). Values were imputed in the CURB-65 score (*n* = 41), Charlson Comorbidity Index (*n* = 2), and CRP (*n* = 11) due to missing information. Overall, the median (IQR) length of hospital stay was 5.5 (3.4–8.5) days, and 2.5% were admitted to the ICU, while 8% had severe CAP with no difference between the groups. The overall 30-day, 90-day, and 180-day rehospitalization rates were 23%, 30% and 40%, respectively. The overall mortality rates were 3% within 30 days, 8% within 90-days, and 12% within 180 days after discharge. Of the 57 undernourished patients, 42 (74%) were categorized based on the complementary diagnostic criteria by ESPEN. Among those with a BMI of <18.5 kg/m^2^, 93% had COPD compared to 45% in the other group (*p* = 0.001). Otherwise, no differences were found in characteristics during hospitalization or outcome characteristics (Appendix A). There was no difference in the causes of re-hospitalization between the groups (Appendix A).

### 4.1. Risk of Re-Hospitalization after Discharge from CAP

The Kaplan-Meier estimate (Figure 2A) shows that all groups had similar risks (15–20%) of re-hospitalization within the first 30 days after discharge. However, between 90 and 180 days after discharge, the re-hospitalization rate increased substantially among undernourished patients with approximately 55% compared to almost 40% among well-nourished and obese patients and 30% among overweight patients. Undernourished, overweight, and obese patients had similar risks of 30-day, 90-day and 180-day re-hospitalization to well-nourished patients (Table 3 ). Compared to overweight patients, obese patients had a higher risk of 90-day (HR: 1.8, 95% CI 1.1; 2.9) and 180-day (HR: 1.8, 95% CI 1.2; 2.6) re-hospitalization. Undernourished patients had a similar risk of 30-day, 90-day, and 180-day re-hospitalization to overweight patients (Appendix A).

### 4.2. Risk of Mortality Risk after Discharge from CAP

The Kaplan-Meier estimate (Figure 2B) shows that the mortality rate increased between 30 and 180 days after discharge among undernourished patients. The proportion who died was approximately 10% 30 days after discharge, 25% within 90 days, and 35% within 180 days after discharge. In comparison, the proportion who died among well-nourished, overweight, and obese patients was 0–8% between 30 and 90 days after discharge and 5–15% 180 days after discharge. Compared to well-nourished patients, undernourished patients had higher risk of 90-day mortality with OR (95% CI) of 3.2 (1.1; 9.9). Undernourished and well-nourished patients had similar 30-day and 180-day mortality risks. Compared to well-nourished patients, overweight and obese patients had similar mortality risks, although no obese patients died within 30 days after discharge (Table 3). Compared to overweight patients, undernourished patients had higher risks of 90-day (OR: 5.0, 95% CI 1.5; 17.0) and 180-day (OR: 5.4, 95% CI 1.8; 16.5) mortality. Overweight and obese patients had similar 30-day, 90-day, and 180-day mortality risks (Appendix A).

## 5. Discussion

Using a complementary definition of undernutrition by ESPEN, we investigated whether undernourished, overweight, and obese patients had increased risk of re-hospitalization and post-discharge mortality after hospitalization with CAP. Compared to well-nourished patients, undernourished patients had increased mortality risk, whereas all groups had similar risk of re-hospitalization as well-nourished. Our results highlight that nutritional status at admission play a role in the prognosis of patients hospitalized with CAP.

There was no difference between the groups regarding the following severity characteristics: length of hospital stay, ICU admission, and CURB-65 score. The median length of stay was 5.5 days in our study population, a median length of stay similar to that reported in several other studies [6,32,33,34]. Admission to the ICU was not frequent in our study population with an overall ICU admission of 2.5%, but this number is in accordance with other studies among CAP patients where the prevalence was 0–9.4% [32,33,34]. Notably, we excluded patients who died in hospital, which resulted in a lower ICU admission rate compared to studies that included these patients. Borisov and colleagues also reported no difference in CAP severity between different BMI categories [34], whereas Kim and colleagues found a lower proportion of severe CAP among obese patients compared to patients with a weight within the normal range [6].

The re-hospitalization rate was high in all four groups at any of the time points. As shown in Table 2 and Figure 2, the four groups had similar 30-day re-hospitalization rates. The re-hospitalization rate between 30 and 90 days remained at approximately 20% in the overweight group, whereas the re-hospitalization rate among underweight, well-nourished, and obese patients increased to approximately 30% from 30 to 90 days after discharge. At 180 days after discharge the re-hospitalization rates had increased in all groups, with the highest increase among undernourished patients (approx. 55% re-hospitalized) followed by obese (approx. 45% re-hospitalized), well-nourished (approx. 40% re-hospitalized), and overweight patients (approx. 30% re-hospitalized). Being undernourished, overweight, or obese were not associated with higher risk of re-hospitalization when compared to well-nourished patients. The differences in re-hospitalization rates observed at 90 and 180 days are therefore caused by other factors, such as COPD and age, both of which were associated with re-hospitalization in the adjusted model. The undernourished patients were the oldest and had the highest proportion of COPD. Therefore, COPD is likely an essential contributing factor for 30-day, 90-day, and 180-day re-hospitalization. CAP is a frequent cause of hospital admission among individuals with COPD. Studies have reported that 3–20% of individuals with COPD require a minimum of one annual hospitalization [35], with exacerbation as a common cause of hospitalization, and the frequency of annual hospitalizations increases with disease progression [36]. Considering the primary causes of re-hospitalization, there was no difference between the groups. Overall, the main cause of re-hospitalization was respiratory problems at 57% (pneumonia in 27.1% and other pulmonary causes in 29.9% whereas 8.5% were caused by cardiovascular problems, and 36.4% were caused by other problems (e.g., pain, anemia, other infections, and behavioral changes). There is limited research focusing on obesity and the risk of re-hospitalization in CAP. Fusco and colleagues investigated whether morbid obesity was associated with 30-day re-hospitalization and found no difference in risk between patients with a BMI of 18.5–29.9 kg/m^2^ and morbid obese (BMI ≥ 40 kg/m^2^) patients [37]. We also found no difference in risk of re-hospitalization within 30 days after discharge between well-nourished and obese patients. Fusco and colleagues reported a proportion of re-hospitalized patients of 12% (BMI between 18.5–39.9 kg/m^2^) [37], which was lower than the proportion found in our study, where 24% of the well-nourished and obese patients and 19% of the overweight patients were re-hospitalized within 30 days after discharge. Fusco and colleagues only focused on short-term re-hospitalization, and they did not compare the risk of 30-day re-hospitalization between obese patients and patients with a normal weight [37]. In our study population, obese patients had an 80% higher risk of 90-day and 180-day re-hospitalization compared to overweight patients. One explanation might be that obesity may worsen symptoms caused by an infection such as CAP. Furthermore, it is possible that obese individuals are more likely to reach out to their general practitioner and that general practitioners tend to refer obese patients to the hospital more often than non-obese patients. Both of these behaviors are likely caused by the perceived risk of an increase in health problems among obese individuals [38].

As shown in the Kaplan-Meier estimate, the mortality rate was largely stable and below 15% among well-nourished, overweight, and obese patients up to 180 days after discharge. The mortality rate steadily increased from approximately 10% 30 days after discharge to approximately 30% 180 days after discharge. Undernourished patients had a 3-fold higher 90-day mortality risk compared to well-nourished patients and a 5-fold higher 90-day and 180-day mortality risk compared to overweight patients. Obese patients had similar mortality risk to well-nourished and overweight patients. Kim and colleagues reported that BMI was associated with 30-day, 6-month, and 1-year mortality. A BMI of ≤22.9 kg/m^2^, ≤23.7 kg/m^2^, and ≤24.1 kg/m^2^ was reported as risk factors for 30-day, 6-month, and 1-year mortality, respectively. Additionally, a protective effect was reported for a BMI between 26.6–38.7 kg/m^2^ on 30-day mortality, a BMI between 27.0–49.2 kg/m^2^ on 6-month mortality, and a BMI of ≥27 kg/m^2^ on 1-year mortality [6]. Borisov and colleagues reported no difference in the risk of 30-day mortality: there was no increased risk in underweight patients or decreased risk in obese patients. However, they only investigated differences in 30-day mortality risk. They reported 30-day mortality rates between 2.5–8.7% with the highest mortality rate in those with a BMI of <18.5 kg/m^2^ [34]. A longer follow-up period might have indicated a higher 90-day mortality risk among undernourished patients, as seen in our study and a study by King and colleagues. They reported that being underweight was associated with a 90-day mortality risk, whereas obesity was associated with a decreased risk of 90-day mortality after hospitalization with CAP [39]. Our study did not support the obesity paradox at 90 or 180 days after discharge, which is likely related to our definition of nutritional status, where the complementary definition of undernutrition led to an improved status of the well-nourished patients (the reference group). Regarding 30-day mortality, our study could neither support nor reject the obesity paradox. As no obese patients died within 30 days after discharge, we were unable to provide OR comparing the risk of mortality between well-nourished and obese patients for this time point.

The main strength of our study is the broader categorization of undernutrition suggested by ESPEN, including weight loss combined with either alternative cut-offs for low BMI according to age or for low FFMI according to sex [24]. The group, appointed by ESPEN, suggested that when using only BMI undernutrition will likely be overlooked in many patients, since an unintentional weight loss of ≥5% within three months or ≥10% within 12 months may not lead to nutritional concerns in individuals presenting a BMI within the normal weight range [24]. In addition, evidence suggests that older individuals have a higher optimal BMI than younger individuals. In our study population, only 15 (26%) of the undernourished patients had a BMI of <18.5 kg/m^2^. The remaining 74% of the undernourished patients would likely have no nutritional concerns, since their BMI was within the normal range. Our results support that patients diagnosed with undernutrition based on the complementary diagnostic criteria suggested by ESPEN should also be considered high-risk, as we found no difference in characteristics during hospitalization and outcome characteristics between the two groups. Other complementary tools for the assessment of undernutrition were suggested by the American Society for Parental and Enteral Nutrition (ASPEN) in 2012 and The Global Initiative on Malnutrition (GLIM) in 2018. ASPEN suggested a diagnosis of undernutrition with the presence of two or more of the following characteristics: insufficient food intake, weight loss, loss of muscle mass, loss of subcutaneous fat, localized or generalized fluid accumulation, and diminished functional status (decreased grip strength) [40]. GLIM suggested two diagnostic assessment criteria, phenotypic and etiologic. The phenotypic criteria include unintentional weight loss, low BMI, and reduced muscle mass, whereas the etiologic criteria include reduced food intake/assimilation and disease burden/inflammatory condition. A diagnosis of malnutrition requires a minimum of one phenotypic and one etiologic criterion [41]. Both ASPEN and GLIM include more characteristics than ESPEN. The GLIM criteria are recommended to promote global communication and consensus on the diagnosis of malnutrition. One may argue that it is a limitation using the ESPEN diagnostic criteria instead of GLIM. However, as stated by GLIM, the diagnostic criteria remain unvalidated in different populations and may not be equally useful in all clinical settings. Using reduced food intake, fluid accumulation, and grip strength at admission may not be the most valid measurements in our patient population. Upon hospital admission, many patients with an acute respiratory infection such as CAP have already experienced reduced food intake and are weaker than usual, and fluid accumulation is also common. Notably, potential fluid accumulation may also be a limitation for the assessment of fat-free mass since accumulated water would be added as part of the fat-free mass by the instrument. However, the ESPEN diagnostic criteria provide a simple and useful tool that seems realistic to integrate as part of clinical practice to assess undernutrition in patients with CAP. To our knowledge, this is the first study using this definition for undernutrition among patients hospitalized with CAP. A limitation is the relatively small sample size. In addition to more statistical power in general, a larger sample size could have provided the OR for comparing the risk of 30-day mortality between well-nourished and obese patients. Another limitation is that height was self-reported and not measured with a stadiometer. Furthermore, the informed consent had to be obtained within 24 h after admission. This may be a limitation, since it is possible some patients refused to participate due to confusion about their situation within the first day of admission, and these patients were likely be more severely ill. As nutritional status was defined at admission, we did not take weight change during admission into consideration, because this was not withing the scope of this study. Many patients across different medical wards lose weight (mainly lean body mass) during hospitalization [42], and, among the elderly, it is likely that weight loss continues after discharge. It is therefore expected that that the nutritional status shifted in a few patients during hospitalization. Using only BMI to define obesity could also have limitations, because BMI does not reflect the location and extent of body fat.

## 6. Conclusions

Compared to well-nourished patients, undernourished patients had an increased risk of 90-day mortality, whereas both undernourished and obese patients had similar risks of re-hospitalization to well-nourished patients. These results highlight that nutrition is an important focus area in the hospital setting and after discharge among patients hospitalized with CAP. Undernourished patients are high-risk patients, and our results indicate that in-hospital screening of undernutrition can identify patients at mortality risk. Studies are needed to investigate whether nutritional therapy after hospitalization with CAP would improve survival.

## Figures and Tables

**Figure 1 nutrients-14-04906-f001:**
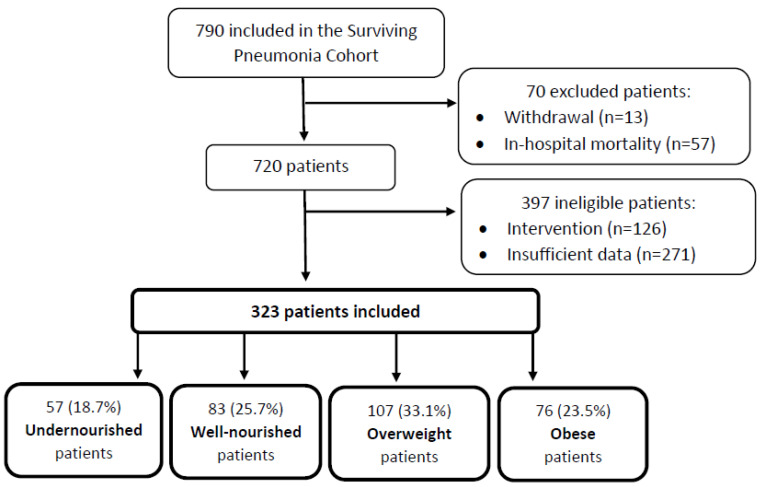
Flow of inclusion in the study.

**Figure 2 nutrients-14-04906-f002:**
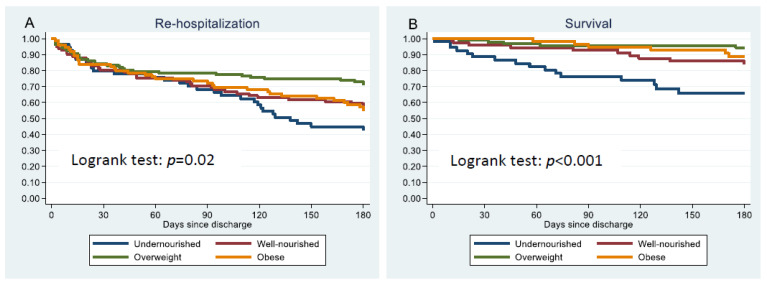
Kaplan-Meier estimates of probability of not having a re-hospitalization (**A**) and survival (**B**) among undernourished, well-nourished, and obese patients after discharge from hospitalization with community acquired-pneumonia.

**Table 1 nutrients-14-04906-t001:** Diagnostic criteria for undernutrition by the European Society for Clinical Nutrition and Metabolism.

Undernutrition Criteria 1	Undernutrition Criteria 2
BMI < 18.5 kg/m^2^	Unintentional weight loss of 5% within 3 months or 10% within 12 months combined with either (1) BMI < 20 kg/m^2^ if <70 years of age or BMI < 22 kg/m^2^ if ≥70 years (2) FFMI defined as <15 kg/m^2^ in females and <17 kg/m^2^ in males

**Table 2 nutrients-14-04906-t002:** Characteristics during hospital admission and outcome measures among undernourished, well-nourished, overweight, and obese patients hospitalized with community-acquired pneumonia.

	Total (*n* = 323)	Undernourished (*n* = 57)	Well-Nourished (*n* = 83)	Overweight (*n* = 107)	Obese (*n* = 76)
Age, years	69.8±13.8	75.4 ± 13.0	69.5±14.4	69.7±12.2	66.0±13.8
Female sex	156 (48.3)	34 (59.7)	38 (45.8)	43 (40.2)	41 (54.0)
BMI, kg/m^2^	26.7±6.1	19.7±2.6	22.9± 1.5	27.4±1.4	35.1±4.6
Fat-free mass index, kg/m^2^	18.0±2.9	15.0 ±1.6	17.0±1.9	18.6±1.8	20.7±3.2
**Comorbidities**					
Charlson Comorbidity index	4.3±2.5	5.2±2.2	4.6 ±2.9	3.9±2.2	3.9 ± 2.6
**Laboratory characteristics**					
C-reactive protein, mg/L *	97.2 (35.7-160.2)	127.8 (53.2-172.3)	107.7 (22.1-158.4)	89.9 (44.7-163.8)	79.7 (28.3-143.1)
**Severity characteristics**					
CURB-65 index					
Mild	85 (26.3)	8 (14.0)	21 (25.3)	31 (29.0)	25 (32.9)
Moderate	211 (65.3)	41 (71.9)	57 (68.7)	69 (64.5)	44 (57.9)
Severe	27 (8.4)	8 (14.0)	5 (6.0)	7 (6.5)	7 (9.2)
Length of stay, days	5.5 (3.6-8.3)	5.5 (3.6-11.2)	5.3 (4.2-8.2)	5.3 (3.2-9.1)	5.7 (3.6-8.3)
ICU admission	8 (2.5)	0 (0.0)	4 (4.9)	3 (2.8)	1 (1.4)
**Outcomes**					
**Re-hospitalization**					
30-day	74 (22.9)	16 (28.1)	20 (24.1)	20 (18.7)	18 (23.7)
90-day	97 (30.0)	20 (53.1)	27 (32.5)	24 (22.4)	26 (34.2)
180-day	129 (39.9)	30 (52.6)	34 (41.0)	31 (29.0)	34 (44.7)
**Mortality**					
30-day	11 (3.4)	7 (12.3)	3 (3.6)	1 (0.9)	0 (0.0)
90-day	25 (7.7)	13 (22.8)	5 (6.0)	4 (3.7)	3 (4.0)
180-day	38 (11.6)	17 (29.8)	10 (12.1)	5 (4.7)	6 (7.9)

Data shown as mean ± SD, median (IQR), or *n* (%). * CRP Levels of ≤3 mg/L was determined to be within the normal range.

**Table 3 nutrients-14-04906-t003:** Risk of re-hospitalization and mortality within 30, 90, and 180 days after discharge among undernourished, well-nourished (reference), overweight, and obese patients hospitalized with community-acquired pneumonia.

	Model 1 (Unadjusted)		Model 2 (Adjusted)	
Re-Hospitalization	HR (95% CI)	*p*-Value	HR (95% CI)	*p*-Value
**30-day**				
Undernourished	1.2 (0.7; 2.0)	0.63	1.0 (0.5; 1.7)	0.89
Well-nourished	Ref.		Ref.	
Overweight	0.8 (0.4; 1.3)	0.36	0.8 (0.5; 1.4)	0.40
Obese	1.0 (0.6; 1.7)	0.96	1.1 (0.6; 1.9)	0.70
**90-day**				
Undernourished	1.1 (0.7; 1.7)	0.79	0.9 (0.5; 1.4)	0.54
Well-nourished	Ref.		Ref.	
Overweight	0.7 (0.4; 1.1)	0.09	0.7 (0.4; 1.1)	0.11
Obese	1.1 (0.7; 1.6)	0.82	1.2 (0.8; 1.9)	0.40
**180-day**				
Undernourished	1.2 (0.9; 1.8)	0.19	1.0 (0.7; 1.5)	0.88
Well-nourished	Ref.		Ref.	
Overweight	0.7 (0.5; 1.0)	0.06	0.7 (0.5; 1.0)	0.08
Obese	1.1 (0.8; 1.6)	0.63	1.3 (0.9; 2.0)	0.20
**Mortality**	**OR (95% CI)**	***p*-value**	**OR (95% CI)**	***p*-value**
**30-day**				
Undernourished	3.7 (0.9; 15.1)	0.07	2.2 (0.5; 9.7)	0.29
Well-nourished	Ref.		Ref.	
Overweight	0.3 (0.02; 2.5)	0.24	0.3 (0.03; 2.7)	0.27
Obese *	-	-	-	-
**90-day**				
Undernourished	4.6 (1.5; 13.8)	0.01	3.2 (1.0; 9.9)	0.049
Well-nourished	Ref.		Ref.	
Overweight	0.6 (0.2; 2.3)	0.47	0.6 (0.2; 2.5)	0.52
Obese	0.6 (0.1; 2.8)	0.55	0.9 (0.2; 3.9)	0.86
**180-day**				
Undernourished	3.1 (1.3; 7.4)	0.01	2.0 (0.8; 5.1)	0.15
Well-nourished	Ref.		Ref.	
Overweight	0.4 (0.1; 1.1)	0.07	0.4 (0.1; 1.2)	0.09
Obese	0.6 (0.2; 1.8)	0.39	0.8 (0.3; 2.5)	0.72

* none died. Cox regression was fitted for re-hospitalization within 30, 90, and 180 days after discharge with mortality as competing event. Logistic regression was fitted for mortality within 30, 90, and 180 days after discharge. Estimates shown are hazard ratios (HR) and odds ratios (OR) with corresponding 95% confidence intervals (CI) and *p*-value. Model 1: unadjusted. Model 2: adjusted for age, sex, and COPD.

## Data Availability

The data set used for the current study is not published. Though by reasonable request to the corresponding author, the data set can be accessed in a pseudonymized form.

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
