# Peer review of "Are Undernutrition and Obesity Associated with Post-Discharge Mortality and Re-Hospitalization after Hospitalization with Community-Acquired Pneumonia?"

_nutrients, 2022, doi:10.3390/nu14224906_

Round 1

Reviewer 1 Report

Abstract: Work to improve the conclusion of abstract.

Introduction: Remove content from 50-53 " with high prevalence to unintentional weight loss". I suggest you write this "Individuals with chronic diseases, such as cancer, chronic obstructive pulmonary disease (COPD), and cardiovascular diseases are at increased risk poor CAP-related outcomes . . .Then present what could be the poor-CAP related outcomes

Remove content from 54-56 "Obesity is  . . . type to diabetes"

Repharase this word: "Paradoxically, in patients with chronic diseases, e.g., patients with different cardiovascular diseases (17, 18), type 2 diabetes (19), but also in patients hospitalized with CAP, obesity has been associated with decreased mortality and better outcomes (20), a phenomenon called the “obesity paradox”." to "On the other hand, obesity has been associated with decreased mortality and better outcomes, even in patient with chronic diseases, i.e., CVD, diabetes, and CAP. 

My suggestion isjust highlight CAP rather than CVD and diabetes everywhere.

Methodology:

Section 2.2 describe CURB 65 in detail.

The definition of underweight is confusing. I suggest you present each criteria in table form

29.9 kg/m2 is overweight and i think it is part of obesity not a well-nourished status. Based on this I suggest you to redo the analysis of your data. I read whole manuscript, I suggest you describe how many you undernourished patient you recived using BMI, how much using weight loss within 3 and 12 months using the scales.

What is the importance of C reactive protein and what is its normal value and then compare it with CAP patients

Result: 

Charslon Morbidity index was not discussed anywhere in method section.

Remove n=294 from the headings inside the table 1, i.e, rehospitalization and mortality after discharge

Merge Table 1 and Table 2. Add a new column total and merge. Undernourished, Well-Nourished, Overnourished and Total.

Put table 3 in your supplementary file

There is no need to discuss insignificant findings in the text. Moreover, no need to discuss unadjusted odds.

I have closely looked into the adjusted and unadjusted model and I found that there is no difference in rehospitalization between any group within 30 days, but within 180 days there is an increased in HR among malnourished pt. So I suggest you merge both undernutrition and overnutrition content into one paragraph rather than 3 to 4. I like your work but organization is not good. 

The message is undernutrition is associated with high odds of mortality and rehospitalization but overnutrition is just associated with rehospitlization. 

Discussion: This need to be improved and no headings inside the discussion section

Sample size is not strength rather it seems a limitation to me. if it is strength then present how it is derived? Is it actually representing the study population? height was not measured this is another limitation.

Author Response

General comments:

Thank you for thorough review. We have met almost all of your inquires except your suggestion to include patients with BMI 25-30 into the obese group. Instead, we have categorized the patients into four groups instead of three. See more detailed explanation later. Another reviewer questioned why we look at outcomes up to 180 days after discharge. The argument for this is that we are interested in looking for opportunities for prevention and therefore we find a longer follow-up period than 30 days more appropriate. This comment made us reflect on the most appropriate follow-up time. We therefore decided to add 90 days as follow-up. An intervention often last 8-12 weeks which is the argument for this. In addition, it has been a while since we extracted the data and since we had to redo the analysis anyway, we added some more patients. The sample size is now 323.

The analyses have been redone and the text in the manuscript are also substantially revised.  

Abstract: Work to improve the conclusion of abstract.

Answer: This is revised

Introduction: Remove content from 50-53 " with high prevalence to unintentional weight loss". I suggest you write this "Individuals with chronic diseases, such as cancer, chronic obstructive pulmonary disease (COPD), and cardiovascular diseases are at increased risk poor CAP-related outcomes . . .Then present what could be the poor-CAP related outcomes

Answer: This is now revised in page 2 line 52-55 (clean manuscript)

Remove content from 54-56 "Obesity is  . . . type to diabetes"

Answer: This is now removed

Repharase this word: "Paradoxically, in patients with chronic diseases, e.g., patients with different cardiovascular diseases (17, 18), type 2 diabetes (19), but also in patients hospitalized with CAP, obesity has been associated with decreased mortality and better outcomes (20), a phenomenon called the “obesity paradox”." to "On the other hand, obesity has been associated with decreased mortality and better outcomes, even in patient with chronic diseases, i.e., CVD, diabetes, and CAP. My suggestion is just highlight CAP rather than CVD and diabetes everywhere.

Answer: this is revised in page 2 line 57-60.

Methodology:

Section 2.2 describe CURB 65 in detail.

Answer: This is now described in more details on page 2 and 3 line 97-99.

The definition of underweight is confusing. I suggest you present each criteria in table form

Answer: We agree. This is now revised on page 3 line 130-133.

Table 1. Diagnostic criteria for undernutrition by European Society for Clinical Nutrition and Metabolism

Undernutrition criteria 1

Undernutrition criteria 2

BMI <18.5 kg/m2

Unintentional weight loss of 5% within 3 months or 10% within 12 months combined with either

1)       BMI <20 kg/m2 if <70 years of age or BMI < 22 kg/m2 if ≥70 years

2)       FFMI defined as <15 kg/m2 in females and <17 kg/m2 in males

29.9 kg/m2 is overweight and i think it is part of obesity not a well-nourished status. Based on this I suggest you to redo the analysis of your data. I read whole manuscript, I suggest you describe how many you undernourished patient you recived using BMI, how much using weight loss within 3 and 12 months using the scales.

Answer: The argument for combining BMI 18.5-24.9 and BMI 25 to 29.9 was that it has been suggested that a BMI above the normal range is beneficial for older individuals and individuals with chronic conditions. Though we agree that it is better to remove patients with BMI 25-29.9 from the well-nourished group. We do not find it appropriate combine overweight with obese patients, thus based on your suggestion we decided to categorize patients into 4 groups; undernourished, well-nourished, overweight, and obese.

In the supplementary material we have a table providing information on the two different subcategories within undernutrition and compared baseline and outcome characteristics. There are 15 patients with BMI <18.5 and 42 patients categorized according to the alternative method. This is described in page 5 line 197-200 and in supplementary table 5.  

What is the importance of C reactive protein and what is its normal value and then compare it with CAP patients

Answer: C-reactive protein is a measure of inflammation measured in the hospital setting to determine if the patient has an infection. It is used to determine the grade of inflammation. When you have community-acquired pneumonia, you will have high CRP levels. The normal value is ≤ 3 mg/L and 3-10 mg/L normal to slightly elevated. We have added the normal information of the normal level in table 2 on page 6.

Result: 

Charslon Morbidity index was not discussed anywhere in method section.

Answer: This is now added in the first section of “data collection” on page 3 line 99-102.

Remove n=294 from the headings inside the table 1, i.e, rehospitalization and mortality after discharge

Answer: This is now table 2 and it is now removed.

Merge Table 1 and Table 2. Add a new column total and merge. Undernourished, Well-Nourished, Overnourished and Total. Put table 3 in your supplementary file

Answer: Table 1 and 2 are merged and table 3 are in supplementary material

There is no need to discuss insignificant findings in the text. Moreover, no need to discuss unadjusted odds.

Answer: The discussion is revised accordingly.

I have closely looked into the adjusted and unadjusted model and I found that there is no difference in rehospitalization between any group within 30 days, but within 180 days there is an increased in HR among malnourished pt. So I suggest you merge both undernutrition and overnutrition content into one paragraph rather than 3 to 4. I like your work but organization is not good. 

Answer: We agree. This is revised in the new version of the manuscript.

The message is undernutrition is associated with high odds of mortality and rehospitalization but overnutrition is just associated with rehospitlization.

Answer: The message is now that obese patients have similar risk of re-hospitalization and mortality as well-nourished patients. Undernourished patients have increased risk of 90-day mortality as well-nourished patients. Otherwise, similar risk. Thus, as I mentioned earlier, we da suspicion that those in the overweight group might have better outcomes. I therefore provided an additional table in the supplementary table evaluating HR and OR with overweight patient as reference group. This is explained in the method section on page 4 line 158-159. 

Discussion: This need to be improved and no headings inside the discussion section

Answer: the discussion has been substantially revised according to this comment and also comments by other reviewers e.g., why we used ESPEN and not definition by GLIM or ASPEN to categorize undernourished patients. This and other issues are discussed in the new version. 

Sample size is not strength rather it seems a limitation to me. if it is strength then present how it is derived? Is it actually representing the study population? height was not measured this is another limitation.

Answer: We agree. It is now revised and added.

Reviewer 2 Report

This is an interesting study investigating the nutritional status and post-discharge prognosis in CAP. However, many concerns exist with this manuscript.

1. I believe it would be easier for the reader to understand if you created an outcome section and described the outcome variable. 

2. Why did you use the 180-day readmission rate as an outcome? In general, I believe the 30-day readmission is often used. Please provide the rationale for using the 180-day readmission as the outcome.

3. The current flowchart (Fig 1) makes the mistake of thinking that nutritional status was measured at discharge. Please revise.

4. The cause of readmission should be clarified.

5. Table 2. is over-nourished a mistake? Is it obese correctly?

6. The causal relationship between obesity and increased 180-day readmission rates for pneumonia patients is unclear. Please discuss this thoroughly in the Discussion.

7.It has been suggested in the global community that the GLIM criteria should be used for nutritional assessment; if the GLIM criteria are unavailable, they should be added to the limitation.

8. If the readmission rate is higher for those with obesity, should obese individuals be given nutritional intervention for weight loss? I disagree with the authors' conclusion since obese patients have a lower mortality rate.

Author Response

Overall comment: Another reviewer commented on the categorization and we therefore decided to categorize the patients into four groups instead of three; undernourished, well-nourished, overweight, and obese. The other reviewer also commented on the sample size as a limitation rather than a strength. We agree. Since we had to redo the analysis anyway, we added more patients recruited since first submission. The sample size is now 323.

The analyses have been redone and the text in the manuscript are also substantially revised.

This is an interesting study investigating the nutritional status and post-discharge prognosis in CAP. However, many concerns exist with this manuscript.

  1. I believe it would be easier for the reader to understand if you created an outcome section and described the outcome variable. 

Answer: We have added out come measures in the heading on page 4 line 141 and the collection of the outcome measures is described in line 142-144. If you think it need further explanation let us know.  

  1. Why did you use the 180-day readmission rate as an outcome? In general, I believe the 30-day readmission is often used. Please provide the rationale for using the 180-day readmission as the outcome.

Answer: The argument for this is that we are interested in looking for opportunities for longer-term prevention and not just readmission from their CAP, therefore we find a longer follow-up period more appropriate. This comment made us reflect on the most appropriate follow-up time. Therefore, we decided to add 90 days as follow-up. An intervention often lasts 8-12 weeks which is the argument for this. In addition, it has been a while since we extracted the data and since we had to redo the analysis anyway, we added more patients. The sample size is now 323.

  1. The current flowchart (Fig 1) makes the mistake of thinking that nutritional status was measured at discharge. Please revise.

Answer: this is now revised to “patients included” on page 5.

  1. The cause of readmission should be clarified.

Answer: We have two tables in supplementary material (suppl. Table 6 and 7) explaining primary causes of re-hospitalization including four categories: pulmonary, pneumonia, cardiovascular and other. And it is also explained in the text on page 5 line 201-202 and page 9 line 286-291. 

  1. Table 2. is over-nourished a mistake? Is it obese correctly?

Answer: Yes, it was. Thanks. It is revised.

  1. The causal relationship between obesity and increased 180-day readmission rates for pneumonia patients is unclear. Please discuss this thoroughly in the Discussion.

Answer: There is no longer an increased risk of 180-day readmission among obese patients compared to well-nourished patients. Though obese patients have 80% higher risk of re-hospitalization compared to overweight patients. We do not have the final answer, but we have now discussed it on page 9 line 303-308.   

7.It has been suggested in the global community that the GLIM criteria should be used for nutritional assessment; if the GLIM criteria are unavailable, they should be added to the limitation.

Answer: Thanks for this comment. We agree that this needed to be discussed. This is done on page 10 line 350-373.

  1. If the readmission rate is higher for those with obesity, should obese individuals be given nutritional intervention for weight loss? I disagree with the authors' conclusion since obese patients have a lower mortality rate.

Answer: Compared to “new” well-nourished group, obese patients do not have a higher re-hospitalization risk. Looking at the mortality rates (based on the new categorization), the mortality rate is lower among obese patients at 30 days after discharge. No obese patients died within 30 days. This is now added to the text. Therefore, we could not conduct the logistic regression for this group at this time point. This is discussed on page 11 line 375-377. The 30-day mortality rate was also low among both well-nourished and overweight patients with a proportion 3.6% and 0.9%. At 90-days it is also similar to overweight and obese and at 180 days the mortality rate seemed to be higher among obese than overweight patients, but not significant.

Reviewer 3 Report

This is an  interesting nested study which should be carefully examined by a qualified statistician before publishing. I do not have this qualifications. 

Based on clinical experience my review of  the paper as a reader with interest and training in nutrition ,  may I suggest that the language and clasification criteria be carefully reviewed?  please see ASPEN guidelines from 2015 and those used in this article are not the same  ( which are in themselves in need of renewal). It is also important to notice that the number of persons that were not included due to lack of data is almost half of the sample size .This raises in itself serious issues about the validity of the data. I am attaching the manuscript with yellow highlighting of some of the questionable areas.

Overall , this paper bring into focus the importance of nutrition in all NCCD. thanks

Author Response

Overall comment: Another reviewer commented on the categorization and we therefore decided to categorize the patients into four groups instead of three as undernourished, well-nourished, overweight, and obese. The other reviewer also commented on the sample size as a limitation rather than a strength. We agree. Since we had to redo the analysis anyway, we added some more patients recruited since first submission. The sample size is now 323.In addition, based on comments from another reviewer, we have added a follow-up at 90-days. The argument for this is that we are interested in looking for opportunities for prevention and therefore we find a follow-up period more appropriate for this purpose, since an intervention often last 8-12 weeks. We have also done additional statistical analysis with overweight patients as reference. The additional table for OR and HR is in the supplementary material. 

The analyses have been redone and the text in the manuscript are also substantially revised.

This is an interesting nested study which should be carefully examined by a qualified statistician before publishing. I do not have this qualifications. 

Answer: One of the co-authors (Prof Ritz) is a professor in statistics and epidemiology and has approved all models as presented.

Based on clinical experience my review of the paper as a reader with interest and training in nutrition ,  may I suggest that the language and clasification criteria be carefully reviewed? 

Answer: We have reviewed the classification criteria again. As mentioned earlier we now have the following groups: undernourished, well-nourished (reference), overweight, and obese. The argument for combining BMI 18.5-24.9 and BMI 25 to 29.9 was that it has been suggested that a BMI above the normal range may be beneficial for older individuals and individuals with some chronic conditions. Though we agreed that it was better to remove patients with BMI 25-29.9 from the well-nourished group. We did not find it appropriate combine overweight with obese patients. In the supplementary material we have a table providing information on the two different subcategories within undernutrition and compared baseline characteristics. There are 15 patients with BMI <18.5 and 42 patients categorized according to the alternative method.

please see ASPEN guidelines from 2015 and those used in this article are not the same (which are in themselves in need of renewal).

Answer: Thanks for this comment. Another reviewer mentioned the diagnostic criteria suggested by GLIM in 2018. We are aware that other tools exist for assessment of undernutrition. We agree that this should be discussed in the manuscript. This is done on page 10 line 350-373.

It is also important to notice that the number of persons that were not included due to lack of data is almost half of the sample size. This raises in itself serious issues about the validity of the data. I am attaching the manuscript with yellow highlighting of some of the questionable areas.

Answer: We understand your concern, however, based on demography in those included the population is highly representative for the general patient population admitted with CAP to our hospital. We mainly had to exclude participants due to insufficient data. We have added a table in supplementary material of ineligible and eligible patients.    

Overall, this paper brings into focus the importance of nutrition in all NCCD. thanks

Round 2

Reviewer 1 Report

Respected author,

I have thoroughly read author response and manuscript again and.I feel that you have not made any significant changes especially in my comments. 
My first comment was to improve conclusion of abstract, where you have added some sentences but I think you have not sent paper for proofreading to other authors that’s why I found many repetitions.

In introduction I asked you to add a line “individual with chronic diseases such as cancer, cops, and cc’d are at high risk of poor cap related outcomes “ you added more or less similar sentence. I really appreciate. But the citation is not relevant. I checked citation 10, where all the diseases which you mentions in your manuscript are secondary outcomes measurement, that manuscript is not saying anything which you wrote.

I didn’t find where you explained CURB 65, you just explained to me what is C reactive protein but you didn’t write in your manuscript. Moreover, you didn’t explain Charslon Morbidity index anywhere. 

In short, you just have merged table 1 and table 2 for me, and also removed headings from the discussion section. 

Best,

Author Response

Respected author,

I have thoroughly read author response and manuscript again and.I feel that you have not made any significant changes especially in my comments. 

Answer:

Dear respected reviewer. We are sincerely sorry that you feel this way. We were happy about your thorough review and in our opinion, we changed the manuscript significantly mainly based on your inputs. Based on your additional suggestions, we have further improved the manuscript. We have tried to meet the specific requests in the updated version.  

My first comment was to improve conclusion of abstract, where you have added some sentences but I think you have not sent paper for proofreading to other authors that’s why I found many repetitions.

Answer:

We think the conclusion is improved and it has been reviewed by co-authors. If you still think we need to improve the conclusion in the abstract, we’re happy to hear your thoughts. We have revised it a little in this version also:

Page 1 line 43-38: In conclusion, among patients with CAP, undernutrition was associated with increased risk of mortality. Undernourished patients are high-risk patients, and our results indicate that in-hospital screening of undernutrition should be implemented to identify patients at mortality risk. Studies are needed to investigate if nutritional therapy after hospitalization with CAP improve survival.

In introduction I asked you to add a line “individual with chronic diseases such as cancer, cops, and cc’d are at high risk of poor cap related outcomes “ you added more or less similar sentence. I really appreciate. But the citation is not relevant. I checked citation 10, where all the diseases which you mentions in your manuscript are secondary outcomes measurement, that manuscript is not saying anything which you wrote.

Answer: We agree. Citation 10 is now removed, and we have added a new one as replacement (page 2 line 56).

I didn’t find where you explained CURB 65

Answer: In the submitted version, we had explained CURB-65 on page 2-3 in line 94-99. However, since you asked for a more detailed description, we have changed it and added a few more details and hope you find the description in te updated version sufficient. The description is changed from:

“Initial disease severity was assessed using the CURB-65 score including the following five parameters: Confusion, Urea (>7mmol/l), respiratory rate ≥30/min), blood pressure (systolic <90 mm Hg or diastolic ≤60 mm Hg), and age (≥65 years). Pneumonia severity was assessed according to the presence of any of the parameters as mild (0-1), moderate (2), or severe (3-5) (25)”.

To

Page 2-3 line 95-100: “Pneumonia severity was assessed using the CURB-65 score – a tool to stratify patients according to mortality risk upon hospital admission. The CURB-65 score includes the following five parameters: Confusion, Urea (>7mmol/l), respiratory rate ≥30/min), blood pressure (systolic <90 mm Hg or diastolic ≤60 mm Hg), and age (≥65 years). One point is provided by presence of any of the parameters and used to categorize pneumonia severity as mild (0-1), moderate (2), or severe (3-5) (25)”.    

you just explained to me what is C reactive protein but you didn’t write in your manuscript.

Answer: We have added an explanation regarding C-reactive protein to the manuscript on page 4 line 149-152.

C-reactive protein (CRP) is a measure of systemic inflammation used as a supporting biomarker when diagnosing infection as well as a response marker to monitor changes during treatment. We determined levels of ≤ 3 mg/L to be within the normal range and levels 3-10 mg/L to be slightly elevated.  

Moreover, you didn’t explain Charslon Morbidity index anywhere. 

Answer: An explanation is now added on page 3 line 104-111).

The Charlson comorbidity index is a tool developed to predict mortality by weighing multiple comorbidities; this includes myocardial infarction, congestive heart failure, peripheral vascular disease, cerebrovascular disease, dementia, chronic pulmonary disease, rheumatologic disease, peptic ulcer disease, mild liver disease, diabetes with or without chronic complications, hemiplegia or paraplegia, renal disease, any malignancy, moderate or severe liver disease, metastatic solid tumor, HIV/AIDS. Each condition provides a score of 1, 2, 3, or 6 depending on their associated risk of mortality.  

In short, you just have merged table 1 and table 2 for me, and also removed headings from the discussion section. 

We may have misunderstood your prior suggestions. Though, as mentioned in the beginning the changes to the manuscript was greatly revised based on your inputs. In our opinion - in addition to the above mentioned - the following changes were based on your comments:

  • The definition of underweight is confusing. I suggest you present each criteria in table form à This was added based on your comment
  • Section 2.2 describe CURB 65 in detail à this was described in more details based on your comment. It is now revised further, and we hope you find the new version sufficient.
  • 9 kg/m2 is overweight and i think it is part of obesity not a well-nourished status. Based on this I suggest you to redo the analysis of your data. I read whole manuscript, I suggest you describe how many you undernourished patient you recived using BMI, how much using weight loss within 3 and 12 months using the scales. à changing the categorization based on your inputs! And there is a supplementary table that describe how many patients that were categorized as undernourished by the to different criteria.
  • I have closely looked into the adjusted and unadjusted model and I found that there is no difference in rehospitalization between any group within 30 days, but within 180 days there is an increased in HR among malnourished pt. So I suggest you merge both undernutrition and overnutrition content into one paragraph rather than 3 to 4. I like your work but organization is not good à Organized discussion based on your comments
  • Sample size is not strength rather it seems a limitation to me. if it is strength then present how it is derived? Is it actually representing the study population? height was not measured this is another limitation àThis is also changed according to your inputs
  • Discussion: This need to be improved and no headings inside the discussion section à We have changed the headings as you requested. In our opinion, we have also revised the discussion significantly. We think the discussion is improved as you requested. If you still think we need to improve the discussion, we’re happy to hear your thoughts.

In addition, the manuscript has been proofread by a native English Proofreader from the company Scribbr. We have added the document from the proofreading company – where the inputs from the proofreader are shown as track changes. The clean document is revised according to these suggested changes. The document with highlights does not have these changes.     

Best,

Reviewer 2 Report

I consider that sufficient corrections were made in response to the comments.

Author Response

There were no comments 

Reviewer 3 Report

1.Please could you do an English language proofing?

2.Please not that the terms you are using " diabetes paradox" are not in any way discussed for the reader and more importantly the term is used in a way that seems to me contradictory to the original meaning of the term . Please take a look at this relevant article :Ahuja, K. R., et al. (2021). "Takotsubo syndrome: Does "Diabetes Paradox" exist?" Heart Lung 50(2): 316-322.

BACKGROUND: Previous small-scale studies have reported conflicting findings regarding the prevalence of diabetes mellitus (DM) and its association with outcomes in patients with takotsubo syndrome (TTS) OBJECTIVE: We sought to assess the prevalence of DM and its association with outcomes in TTS patients. METHODS: Nationwide inpatient sample (NIS) was queried to extract patient information from January 1, 2009 to September 30, 2015. Propensity score matching (PSM) was done to compare mortality and other in-hospital outcomes. RESULTS: A total of 40,327 hospitalizations for TTS were included. The prevalence of DM in TTS was 19.3% vs 23.1% (p-value < 0.01) in patients without TTS in the NIS from 2009 to 2015. In the PSM cohort, there was no difference in in-hospital mortality (1.1% vs 1.4%; p = 0.76), stroke (1.2% vs 0.9%; p = 0.09), cardiogenic shock (3.7% vs 3.9%; p = 0.61), cardiac arrest (1.2% vs 1.2%; p = 0.94), ventricular arrhythmias (3.7% vs 3.3%; p = 0.23), circulatory support (2.1% vs 1.8%; p = 0.17), and invasive mechanical ventilation (4.9% vs 4.7%; p = 0.54) in TTS patients with versus without diabetes. In sub-group analysis, diabetes with chronic complications patients were found to have lower mortality (0.7% vs 2.0%; p = 0.04) compared to patients without diabetes and those with uncomplicated diabetes (0.6% vs 2.6%; p = 0.002). CONCLUSIONS: Prevalence of DM was lower in TTS in comparison to patients without TTS. In addition, complicated DM patients were found to have lower in-hospital mortality. Further studies are needed to assess the mid and long-term outcomes of DM with and without chronic complications in .

Author Response

1.Please could you do an English language proofing?

Answer: The manuscript has now been proof read by the a native English Proofreader from the company Scribbr. We have added the document from the proofreading company – where the inputs from the proofreader are shown as track changes. The clean document is revised according to these suggested changes. The document with highlights does not have these changes.    

2.Please not that the terms you are using " diabetes paradox" are not in any way discussed for the reader and more importantly the term is used in a way that seems to me contradictory to the original meaning of the term . Please take a look at this relevant article :Ahuja, K. R., et al. (2021). "Takotsubo syndrome: Does "Diabetes Paradox" exist?" Heart Lung 50(2): 316-322.

 BACKGROUND: Previous small-scale studies have reported conflicting findings regarding the prevalence of diabetes mellitus (DM) and its association with outcomes in patients with takotsubo syndrome (TTS) OBJECTIVE: We sought to assess the prevalence of DM and its association with outcomes in TTS patients. METHODS: Nationwide inpatient sample (NIS) was queried to extract patient information from January 1, 2009 to September 30, 2015. Propensity score matching (PSM) was done to compare mortality and other in-hospital outcomes. RESULTS: A total of 40,327 hospitalizations for TTS were included. The prevalence of DM in TTS was 19.3% vs 23.1% (p-value < 0.01) in patients without TTS in the NIS from 2009 to 2015. In the PSM cohort, there was no difference in in-hospital mortality (1.1% vs 1.4%; p = 0.76), stroke (1.2% vs 0.9%; p = 0.09), cardiogenic shock (3.7% vs 3.9%; p = 0.61), cardiac arrest (1.2% vs 1.2%; p = 0.94), ventricular arrhythmias (3.7% vs 3.3%; p = 0.23), circulatory support (2.1% vs 1.8%; p = 0.17), and invasive mechanical ventilation (4.9% vs 4.7%; p = 0.54) in TTS patients with versus without diabetes. In sub-group analysis, diabetes with chronic complications patients were found to have lower mortality (0.7% vs 2.0%; p = 0.04) compared to patients without diabetes and those with uncomplicated diabetes (0.6% vs 2.6%; p = 0.002). CONCLUSIONS: Prevalence of DM was lower in TTS in comparison to patients without TTS. In addition, complicated DM patients were found to have lower in-hospital mortality. Further studies are needed to assess the mid and long-term outcomes of DM with and without chronic complications in .

Answer: We are not discussing a diabetes paradox, but an obesity paradox in patients with diabetes. Meaning that obese patients with diabetes may have better outcomes (=improved survival) compared to patients with normal weight. I hope this is now clear. Otherwise, if it is confusing we can renmove cardiovascular diseases and diabetes from this sentence:

From:

“On the other hand, obesity has been associated with decreased mortality and better outcomes, even in patients with chronic diseases such as cardiovascular diseases (18, 19), type 2 diabetes (20), but also in patients hospitalized with CAP (21), a phenomenon called the “obesity paradox”.”

To:

“On the other hand, obesity has been associated with decreased mortality and better outcomes in patients hospitalized with CAP (21), a phenomenon called the “obesity paradox”.”
